# The Sensory Quality Improvement of Citrus Wine through Co-Fermentations with Selected Non-*Saccharomyces* Yeast Strains and *Saccharomyces cerevisiae*

**DOI:** 10.3390/microorganisms8030323

**Published:** 2020-02-26

**Authors:** Lanlan Hu, Rui Liu, Xiaohong Wang, Xiuyan Zhang

**Affiliations:** 1College of Food Science and Technology, Huazhong Agricultural University, Wuhan 430070, China; Hulanlan@webmail.hzau.edu.cn (L.H.); Liurui@webmail.hzau.edu.cn (R.L.); Wxh@webmail.hzau.edu.cn (X.W.); 2Hubei International Scientific and Technological Cooperation Base of Traditional Fermented Foods, College of Food Science and Technology, Huazhong Agricultural University, Wuhan 430070, China

**Keywords:** Citrus wine, Non-*Saccharomyces* yeast strains, Co-fermentation, Sensory quality

## Abstract

Co-fermentation of selected non-*Saccharomyces* yeast strain with *Saccharomyces cerevisiae* is regarded as a promising approach to improve the sensory quality of fruit wine. To evaluate the effects of co-fermentations between the selected non-*Saccharomyces* yeast strains (*Hanseniaspora opuntiae*, *Hanseniaspora uvarum* and *Torulaspora delbrueckii*) and *S. cerevisiae* on the sensory quality of citrus wine, the fermentation processes, the chemical compositions, and the sensory evaluations of citrus wines were analyzed. Compared with those of *S. cerevisiae* fermentation, co-fermentations produced high sensory qualities, and *S. cerevisiae*/*H. opuntiae* co-fermentation had the best sensory quality followed by Sc-Hu and Sc-Td co-fermentations. Additionally, all the co-fermentations had a lower amount of ethanol and total acidity, higher pH value, and higher content of volatile aroma compounds, especially the content of higher alcohol and ester compounds, than those of *S. cerevisiae* fermentation. Therefore, co-fermentations of the non-*Saccharomyces* yeast strains and *S. cerevisiae* could be employed to improve the sensory quality of citrus wines. These results would provide not only methods to improve the sensory quality of citrus wine, but also a valuable reference for the selection of non-*Saccharomyces* yeast strains for fruit wine fermentation.

## 1. Introduction

Citrus is one of the most abundant fruit crops in China, with a production of 8.56 million tons, accounting for 11.8% of the global production in 2017 (FAO, 2017). Mandarin fruit cv. Ponkan (*Citrus reticulata* Blanco cv. Ponkan) is widely cultivated in Asian countries for its high quality [1]. Ponkan has a thin skin, and the quality of the fruit during storage at room temperature is highly prone to deterioration and decay over time. Improving storage performance and reducing fruit decay during storage and transportation are major issues for the citrus industry [2,3]. In addition to being consumed as fresh fruit, citrus can also be processed into citrus wine to extend its shelf life and increase its added value [4]. However, the insufficient research and development of fermented citrus wine, as well as the poor flavor and quality of citrus wine, decreased its competitiveness on the fruit wine market [5].

Yeasts are primarily responsible for the alcoholic fermentation of fruit juice. Industrial fruit wine is usually fermented by using commercial *Saccharomyces cerevisiae* with the advantages of the controllable processes and stable quality products, but its sensory quality is inferior to those of successful spontaneous fermentation [6]. Nowadays, more and more non-*Saccharomyces* yeast strains, such as *Torulaspora delbrueckii, Hanseniaspora uvarum*, *Hanseniaspora vineae*, *Metschnikowia pulcherrima, Lachancea thermotolerans, Pichia kluyveri, Pichia fermentans, Candida zemplinina, Candida stellate, Zygotorulaspora florentina* and *Schizosaccharomyces pombe*, were employed to ferment fruit wine with *S. cerevisiae* to improve the sensory quality of fruit wine by producing low content of volatile acids, high content of aroma compounds and glycerol [7,8,9,10,11,12,13,14,15]. However, some non-*Saccharomyces* yeast strains might produce undesirable metabolites during fruit wine fermentation, such as acetic acid, aldehyde, volatile phenols, and acetoin etc. [16,17,18], while others might not grow normally under the harsh fermentation environments (poor nutrition, low pH and temperature, high content of ethanol and SO_2_, competition from other microorganisms) [19,20]. All aspects generally considered for non-*Saccharomyces* in grape musts/wines in the current subject research [20,21,22,23]. Therefore, it is important to study the effects of non-*Saccharomyces* yeast strains on the fermentation process and the sensory quality of non-grape fruit wine. Our research group reported that the pure fermentation of *Hanseniaspora opuntiae*, *H. uvarum,* or *T. delbrueckii* displayed positive effects on the sensory quality of citrus wine, while they also had an incomplete ethanol fermentation [24]. Therefore, these non-*Saccharomyces* yeast strains can be used to ferment citrus wine with *S. cerevisiae* which had outstanding sugar consumption ability [6]. However, it is still unknown whether co-fermentation of these three non-*Saccharomyces* yeast strains with *S. cerevisiae* could improve the sensory quality of citrus wine with complete ethanol fermentation or not.

To investigate the effects of co-fermentations between the selected non-*Saccharomyces* yeast strains and *S. cerevisiae* on the citrus wine, the fermentation processes, the chemical compositions and sensory evaluations of citrus wines were analyzed. Research results would provide methods to improve the sensory quality of citrus wine and provide references for the selection of non-*Saccharomyces* yeast strains for fruit wine fermentation.

## 2. Materials and Methods

### 2.1. Citrus Juice Preparation

The citrus fruit (*Ponkan*) was harvested from Wuhan citrus plantation (Wuhan city, Hubei province, China) in 2016. The citrus juice (pH 3.36, 11 Brix of total soluble solids, 60.37 g/L of initial sugar, 3.8 g/L of total acidity) was prepared by peeling, crushing and squeezing from fresh citrus, and then pasteurizing for 15 min at 104 °C.

### 2.2. Yeast Strains and Media

*Hanseniaspora opuntiae*, *Hanseniaspora uvarum* and *Torulaspora delbrueckii* were isolated from spontaneously fermented citrus wine and orangeries [5]. *Saccharomyces cerevisiae* was obtained from the Laffort group (Actiflore Cerevisiae, Laffort Co., Bordeaux, France).

Yeasts extract peptone dextrose (YEPD) medium (10.0 g/L yeasts extract, 20.0 g/L peptone, and 20.0 g/L dextrose) was employed to culture the starters.

Wallerstein laboratory nutrient agar (WL) medium (50.0 g/L dextrose, 0.125 g/L MgSO_4_, 5.0 g/L tryptone, 0.022 g/L bromocresol green, 4.0 g/L yeasts extract, 0.0025 g/L FeCl_3_, 0.55 g/L KH_2_PO_4_, 0.0025 g/L MnSO_4_, 0.425 g/L KCl, 15.0 g/L agar and 0.125 g/L CaCl_2_, pH 5.5) was used to differentiate *S. cerevisiae* from non-*Saccharomyces* yeast strains according to the colony morphology [25].

Lysine agar (LYS) medium (10.0 g/L dextrose, 5.0 g/L lysine, 0.1 g/L KH_2_PO_4_, 0.1 g/L MgSO_4_, complex vitamin 0.1 g/L and agar 15.0 g/L, pH 4.8) was used to count the non-*Saccharomyces* yeast strain cells during co-fermentations.

### 2.3. Laboratory-Scale Fermentation of Citrus Wine

After adjusted to 210 g/L sugar and 50 mg/L SO_2_ by exogenous addition of 104.741 g sucrose and 0.566 mL saturated sulfurous acid solution, respectively, 700 mL citrus juice was fermented with starters of *S. cerevisiae/H. uvarum* (Sc-Hu), *S. cerevisiae/T. delbrueckii* (Sc-Td) or *S. cerevisiae/H. opuntiae* (Sc-Hop) in 1.0 L sterile bottles at 25 °C. The co-fermentations were performed with co-cultures of 10^7^ CFU/mL non-*Saccharomyces* yeast strain and 10^6^ CFU/mL *S. cerevisiae* with sequence inoculation strategy. The sequence inoculation strategy meant that *S. cerevisiae* was inoculated into citrus juice 24 h later than *H. opuntiae* and *T. delbrueckii*, or 72 h later than *H. uvarum.* Pure fermentation of *S. cerevisiae* with 10^6^ CFU/mL inoculation was used as a control. The fermentation process of citrus wine was monitored by analyzing the residual sugar concentration and yeast cell counts daily. Residual sugar concentration of citrus wine was assessed by dinitrosalicylic acid (DNS) method according to the International Organization of the Vine and Wine (OIV, 2005). Yeast cell count was determined by successive dilution method on the WL medium and LYS medium. All reagents were obtained from Sigma-Aldrich (Sigma-Aldrich, Shanghai, China).

### 2.4. Physicochemical Analysis of Citrus Wine

The content of residual sugar, total acidity (expressed as g/L of malic acid) and volatile acid (expressed as g/L of acetic acid) of citrus wine were analyzed by the International Organization of the Vine and Wine (OIV, 2005). Bromothymol blue titration method was used to analyze the total acidity of citrus wine. Distillation and titration method was applied to evaluate the volatile acid of citrus wine. Rapid oxidation of potassium permanganate method was employed to detect the alcohol content of citrus wine. The pH value of citrus wine was determined by pH meter (Mettler-Toledo, Shanghai, China). All experiments were determined in triplicate.

### 2.5. HS-SPME/GC-MS Analysis of Volatile Aroma Compounds

The volatile aroma compounds were extracted by headspace solid-phase microextraction (HS-SPME) method with a 50/30 μm divinylbenzene/carboxen/polydimethylsiloxane (DVB/CAR/PDMS) fiber (Supelco, Bellefonte PA, USA). The Agilent 6890N gas chromatography (Agilent 6890N, Agilent Technologies Inc., Shanghai, China) on an HP-5 capillary column (30 m × 0.32 mm × 0.25 μm) coupled to an Agilent 5975B mass spectrometer was used to analyze the extracted volatile aroma compounds. The extraction, analysis, and identification of volatile aroma compounds were conducted as described by Hu et al. [24]. The odor active value (OAV), calculated as the ratio between the concentration of flavor compound to its odor threshold (OT), was used to obtain odor patterns. Volatile aroma compounds with OAV≥1 were considered as odor-active compounds [26].

### 2.6. Sensory Evaluation of Citrus Wine

The sensory evaluation of citrus wine was conducted by a trained panel consisting of nine panelists (five females and four males) from Huazhong Agricultural University. The sensory evaluation of citrus wine was performed as described by [24].

### 2.7. Statistical Analyses

One-way ANOVA and Duncan test of all indicated significant differences were conducted by SPSS 19.0 (SPSS Inc., Chicago, IL, USA). Principal component analysis (PCA) was performed to identify the most influential volatile aroma compounds in different fermentations by SIMCA-P 14.1 (Umetrics AB, Umea, Sweden). Hierarchical cluster analysis and heat map visualization of volatile aroma compounds with the Z-score standardization in different fermentations were analyzed by MultiExperiment Viewer 4.9.0 (TIGR, America).

## 3. Results and Discussions

### 3.1. Growth Kinetics and Sugar Consumption Kinetics of Yeast Strains during Fermentations

As charted in Figure 1, these non-*Saccharomyces* yeast strains shortly adapted to the fermentation environment, then grew normally within 4-6 days and reached their maximum biomass of 1.0 × 10^8^ CFU/mL. However, the growth of *H. uvarum* was immediately inhibited by *S. cerevisiae* with sharp decrease of biomass during fermentation (Figure 1A). The growth inhibition phenomenon of non-*Saccharomyces* yeast strains were also reported in other studies [27,28]. Additionally, the early inoculation of these non-*Saccharomyces* yeast strains negligibly affected the maximum biomass of *S. cerevisiae* (1.4 × 10^8^–1.9 × 10^8^ CFU/mL) in co-fermentations compared with that of pure *S. cerevisiae* fermentation (2.3 × 10^8^ CFU/mL) (Figure 1A–D). These results indicated the selected non-*Saccharomyces* yeast strains and *S. cerevisiae* could grow normally during their co-fermentation with the exception of *H. uvarum* which was inhibited by *S. cerevisiae.* The inhibition of non-*Saccharomyces* yeast strains was probably ascribed to the fierce nutrients competition or killer factors from *S. cerevisiae* [29].

### 3.2. Physicochemical Parameters of Citrus Wine

As presented in Table 1, co-fermentations of non-*Saccharomyces* yeast strains and *S. cerevisiae* contained lower concentrations of ethanol (9.74%–10.24%) and total acidity (7.68–8.63 g/L) than pure *S. cerevisiae* fermentation did (11.29% and 9.44 g/L). The Sc-Hu co-fermentation possessed a lower concentration of residual sugar (1.38 g/L), while others contained a higher level of residual sugar (2.03–2.81 g/L) than pure *S. cerevisiae* fermentation did (2.03 g/L), which might be caused by different sugar consumption abilities of these yeast strains [30,31]. The pH value of co-fermentations (3.37–3.47) increased compared with that of pure *S. cerevisiae* fermentation (3.35), which was also found in co-fermented bilberry wine [32]. Additionally, the volatile acid concentrations in co-fermentations (0.11–0.12 g/L) with the exception of Sc-Td co-fermentation (0.15 g/L) displayed insignificant difference from that of pure *S. cerevisiae* fermentation (0.11 g/L), while other researchers reported that co-fermentations had a lower content of volatile acid than pure *S. cerevisiae* fermentation had [8,33]. The different research results in volatile acid content might be caused by different fruit juice, fermentation strategies or starters. These results indicated these co-fermentations produced citrus wine with lower content of ethanol and total acidity, higher pH value and with no significant difference in volatile acid concentration.

### 3.3. Volatile Aroma Compounds in Citrus Wines

Forty-eight kinds of volatile aroma compounds in citrus wines, including 8 kinds of higher alcohols, 6 kinds of fatty acids, 24 kinds of esters and 10 kinds of other compounds, were detected. Compared with the *S. cerevisiae* fermentation, co-fermentations of these non-*Saccharomyces* yeast strains and *S. cerevisiae* produced higher content of volatile aroma compounds, and Sc-Td co-fermentation produced the maximum amount of volatile aroma compounds (819.46 mg/L), followed by Sc-Hop (689.32 mg/L), Sc-Hu (519.97 mg/L) and Sc fermentations (187.58 mg/L). Moreover, co-fermentations produced higher content of higher alcohol, acetate and terpene, and lower content of volatile fatty acids with the exception of Sc-Td fermentation (Table 2).

#### 3.3.1. Higher Alcohols

Higher alcohols contributed positively to the “fresh fruity”, “vegetal” notes, and aroma complexity in young red wine [34]. Higher alcohols could react with organic acids to form esters with a pleasant flavor, while excessive levels (≥500 mg/L) may lead to unpleasant flavor in alcohol beverage [35]. As tabulated in Table 2 and Table 3, co-fermentations produced higher concentrations of higher alcohols (235.52 mg/L–580.72 mg/L) than that of pure *S. cerevisiae* fermentation did (87.97 mg/L). The Sc-Td co-fermentation produced the maximum concentration of higher alcohols (580.72 mg/L), followed by Sc-Hop (410.48 mg/L) and Sc-Hu (235.52 mg/L) fermentations. However, excessive concentration of higher alcohols (580.72 mg/L) in Sc-Td co-fermentation might result in undesirable flavor. Previous research also showed that *S. cerevisiae*/*T. delbrueckii* and *S. cerevisiae*/*H. opuntiae* co-fermentations generated large amounts of higher alcohols compared with pure *S. cerevisiae* fermentation did, while the differences in production of higher alcohols among different non-*Saccharomyces* strains were significant [36,37,38].

Among the detected alcohol groups, 1-pentanol and phenylethanol were the active odor compounds (OAV > 1), and the phenylethanol concentration in co-fermentations (183.00 mg/L–306.33 mg/L) was greatly higher than that of pure *S. cerevisiae* fermentation (33.21 mg/L), which was also reported by Sun et al. [33]. Phenylethanol usually presented on pleasant honey, floral aroma (rose petals), and spicy flavor [39,40]. Therefore, a higher amount of phenylethanol in co-fermentations would be positive to the flavor of citrus wines. Besides, the content of 1-pentanol with sweet and vanilla odor was higher in co-fermentation with the exceptions of Sc-Hu fermentation than that of pure *S. cerevisiae* fermentation did, which would strengthen the fruity and balsamic of citrus wine. These results revealed that co-fermentations of the non-*Saccharomyces* yeast strains and *S. cerevisiae* significantly contributed to the biosynthesis of higher alcohols, especially phenylethanol, and the higher alcohols production in fruit wine was determined by non-*Saccharomyces* yeast strain and the fermentation process.

#### 3.3.2. Fatty Acids

As viewed in Table 2 and Table 3, the Sc-Td co-fermentation produced higher concentration of fatty acids (20.04 mg/L) than *S. cerevisiae* fermentation did (2.54 mg/L), while other co-fermentations hardly produced fatty acids (0.00 mg/L-0.16 mg/L). Volatile fatty acids are essential precursors of ester productions that provide fruity aromas to wines [47], but excessive content of fatty acids (≥20 mg/L) in Sc-Td co-fermentation might produce rancid flavor [48,49]. Octatonic acid was the only odor-active fatty acid and presented in Sc-Td co-fermentation (OAV = 40.08) and pure *S. cerevisiae* fermentation (OAV = 4.45). The high content of fatty acids, especially octatonic acid, in Sc-Td co-fermentation was inconsistent with previous report which indicated that co-fermentation of *T. delbrueckii* and *S. cerevisiae* greatly decreased the fatty acids concentration, especially octatonic acid [50], which might be caused by different fruit juice and fermentative process. These results indicated that co-fermentations of the selected non-*Saccharomyces* yeast strains and *S. cerevisiae* have no regular effects on the amount and the kinds of fatty acids in citrus wine, which might be determined by yeast strains, fruit juice, and fermentation process.

#### 3.3.3. Esters

Esters compounds including acetates and ethyl esters are important aroma compounds with a positive contribution to the desired fruit aroma characters in wine [51]. The total ester concentrations in all the co-fermentations (210.52 mg/L-281.61 mg/L) with significant differences (*p* < 0.05) were higher than that of pure *S. cerevisiae* fermentation (96.84 mg/L) (Table 2), which was also found in bilberry wine co-fermented by *T. delbrueckii* and *S. cerevisiae* [39].

As for the acetates, all the co-fermentations also generated higher acetates amount than pure *S. cerevisiae* fermentation did, and Sc-Hu fermentation produced the maximum acetates content (236.82 mg/L), followed by Sc-Hop (199.59 mg/L) and Sc-Td fermentations (126.36 mg/L). Among the acetate compounds, ethyl acetate, isoamyl acetate, and phenethyl acetate were the odor active compounds (OAV > 1) and greatly increased in all the co-fermentations compared with those of *S. cerevisiae* fermentation. Ethyl acetate may improve aroma complexity of wine at low level (approximately 50 mg/L), but it is associated with negative sensory descriptors (nail polish and solvent etc.) at concentrations above 150 mg/L. Therefore, the ethyl acetate concentration in Sc-Td and Sc-Hop fermentations (32.04 mg/L and 81.64 mg/L, respectively) would have positive effect on the flavor of citrus wine, while that in Sc-Hu fermentation (153.50 mg/L) might has negative effect on the sensory quality of citrus wine and should be evaluated through sensory evaluation furtherly. However, Mingorance-Cazorla et al. reported that 9.67 mg/L–163.18 mg/L ethyl acetate were also detected from citrus wine, and a little over threshold of ethyl acetate in citrus wine (163.18 mg/L) had no negative effect on its flavor [52]. Isoamyl acetate and phenethyl acetate are recognized as an important flavor compound in wine and contribute to the fruity notes of wine [39]. Moreira et al. [53] and Rojas et al. [54] also revealed that non-*Saccharomyces* yeast, such as *H. uvarum*, could increase the content of isoamyl acetate and the phenethyl acetate in wine. These results suggested that co-fermentations could improve the acetate content, especially the content of ethyl acetate, isoamyl acetate and phenethyl acetates in citrus wine compared with those of pure *S. cerevisiae* fermentation, which would be favorable to the flavor of citrus wine.

As for ethyl esters, Sc-Td and Sc-Hop fermentations displayed distinct advantage in ethyl esters producing (77.17 mg/L and 73.31 mg/L, respectively), while Sc-Hu fermentation (43.99 mg/L) was slightly inferior to *S. cerevisiae* fermentation (66.62 mg/L) (Table 2). However, Hu et al. [55] reported co-culture of *H. uvarum*/*S. cerevisiae* produced more ethyl ester in wine than *S. cerevisiae* did. These differences in producing abilities of ethyl ester among different yeast strains might be caused by their different producing ability of organic acids, because the organic acids were the precursors of ethyl esters [56]. Among the ethyl ester compounds, ethyl caproate, ethyl octanoate and ethyl caprate were the odor active compounds, which were characterized by “pineapple, fruity and floral”, “fruity, pineapple, pear and floral”, and “fruity and fresh”, respectively (Table 3). These results suggested that co-fermentations of the non-*Saccharomyces* yeast strains and *S. cerevisiae* exhibited great potential in ethyl esters compounds production in citrus wine.

The content of other esters in Sc-Td co-fermentation was higher than that of other fermentations. These results suggested that different co-fermentations had no regular effects on other esters content.

#### 3.3.4. Volatile Terpenes, Aldehydes, Ketones, and Phenols

Major volatile compounds in citrus wine were higher alcohols, esters and fatty acids, but the volatile aldehydes, ketones, terpenes and phenols were also identified. Various types of terpene and aldehyde compound, such as D-limonene, linalool, octanal, and decanal etc. have been reported in citrus juice [57,58]. Terpenes are important varietal aroma compounds and can impact on floral aroma of wines [59]. Co-fermentations significantly enhanced the amount and the kinds of terpene compounds in citrus wine (2.12 mg/L–12.98 mg/L) compared with those of pure *S. cerevisiae* fermentation (0.00 mg/L) (Table 3). The level of terpene compounds was also enhanced in wine and mango wine by non-*Saccharomyces* yeast strains [60,61]. Only D-limonene in Sc-Hu and Sc-Hop fermentations and 1-pentene in Sc-Hop fermentation had the odor activities (Table 2). Aldehydes and ketones were slightly produced in Sc-Hu (0.56 mg/L) and Sc-Hop (0.55 mg/L) fermentations, while they were not detected in other fermentations. Only phenylacetaldehyde in Sc-Hu and Sc-Hop fermentation was odor active, and would present on unpleasant green notes [62]. Sc-Td co-fermentation contained higher phenol content (5.20 mg/L) than other fermentations did (0.00 mg/L-0.23 mg/L), and 4-vinyl guaiacol in Sc-Td fermentation was the only odor active phenol compound which would present on coffee, beer, apple aroma (Table 2 and Table 3). Different content of terpenes, aldehydes, ketones, and phenols in different fermentations would take on different variety of flavor characteristics in citrus wines and determined by different yeast strains.

### 3.4. Principal Component Analysis of Volatile Compounds in Citrus Wine

The principal component analysis (PCA) demonstrated the correlation and segregation of odor active compounds (OAV≥1) with fermentations (Figure 2). PCA explained 89.7% of the total variation. The first principal component (PC1) accounted for 54.5% of the total variation, while PC2 explained 35.2%. Sc-Td co-fermentation significantly clustered with several odor active compounds including ethyl caproate, 1-pentanol, ethyl butyrate, 4-vinyl guaicol and octanoic acid, at the upper right corner, while Sc-Hop and Sc-Hu fermentations grouped together with odor active compounds such as benzene acetaldehyde, 1-pentene, D-limonene and ethyl acetate, at the upper left corner. However, *S. cerevisiae* fermentation located at the lower right corner with ethyl octanoate and ethyl caprate. Similarly, Zhang et al. indicated that the *S. cerevisiae* monoculture mainly gathered with ethyl octanoate [11]. These results highlighted that different fermentation strategies produced distinctive odor active compounds and would present on different flavor characteristics in citrus wine.

### 3.5. Hierarchical Cluster Analysis of Volatile Aroma Compounds in Citrus Wine

Hierarchical cluster analysis was used to differentiate the volatile aroma compound profiles of different fermentations. As revealed in Figure 3, the fermentations were classified into three groups including group of Sc-Td co-fermentation, group of Sc-Hu and Sc-Hop fermentations, and group of Sc fermentation. All the detected volatile compounds from fermentations were clustered into four groups and designated as group I, II, III and IV. Meanwhile, The Sc-Td co-fermentation was rich in volatile aroma compounds of group I (ethyl octanoate and ethyl dodecanoate) and group IV (isoamyl acetate, phenylethanol, 3,5-cycloheptatriene, isoamyl formate, ethyl caproate, pentyl octanoate, ethyl butyrate, octanoic acid, 1-pentanol, isolongifolan-8-ol, 2,4-dimethyl-benzoic acid, isopropyl caprylate, ethyl heptanoate, ethyl octadecenoate, pentyl octanoate, propyl 4-phenylbutyrate, pentyl propanate, 1-octene, 1,4-dimethyl-2-vinyl-benzene and 3,4-dimethylphenol). The Sc-Hu and Sc-Hop co-fermentations were abundant in compounds of group II (ethyl acetate, ethyl propionate, 4-methylphenylpropanol, amyl acetate, ethyl cis 4-decenoate, hexyl phenylalanate, 4-ethylbenzoic acid, 2,4-dimethyl-benzoic acid, 4-pentenyl-benzene, D-Limonene and benzenebutanal), and group III (ethyl formate, ethyl undecanoate, ethyl 4-methyl-valerate, amyl formate, 1-pentene and phenylacetaldehyde). However, the volatile compounds from the pure *S. cerevisiae* fermentation belonged to group I (ethyl octanoate, ethyl dodecanoate, ethyl caprate, undecanoic acid, hexanoic acid, dodecanoic acid, and ribitol). These results revealed that different fermentation strategies presented different aroma compound profiles which would take on different flavor complexities and characteristics in citrus wine.

### 3.6. Sensory Evaluations of Citrus Wines

The sensory evaluations of citrus wines were shown in Table 4. Compared with those of pure *S. cerevisiae* fermentation, the sensory evaluation scores of co-fermentations (13.00-15.00) were higher, and the Sc-Hop co-fermentation received the highest one (15.00), followed by Sc-Hu (14.00) and Sc-Td co-fermentations (13.00). More importantly, the clarity, aroma, taste, taste lasting and overall acceptability of co-fermentations were significantly improved. In detail, aroma and taste of Sc-Hop and Sc-Hu fermentations were greatly improved, followed by Sc-Td fermentations. Meanwhile, the appearance, taste lasting attribute and overall acceptability of co-fermentations were strongly enhanced. Previous researches reported that co-fermentations of non-*Saccharomyces* yeast strains, such as *H. opuntiae*, *P. kudriavzevii*, *H. uvarum*, *T. delbrueckii* etc., and *S. cerevisiae* could improve the fruity, floral, and mouth-feeling of wine [63,64,65]. Therefore, *H. opuntiae*, *H. uvarum*, and *T. delbrueckii* could be employed to co-ferment with *S. cerevisiae* to improve the sensory quality of citrus wine. The Sc-Hop fermentation was the best method to produce citrus wine with outstanding aroma, taste, taste lasting, and overall acceptability, followed by fermentations of Sc-Hu and Sc-Td, respectively.

The correlation analysis was employed to study the correlation between sensory evaluation scores and aroma substances. These results showed that the main indicators of sensory evaluation were more significantly related to the esters, terpenes, aldehydes, and ketones (*p* < 0.01) than the other volatile compounds (Table 5). Therefore, the sensory evaluation score of citrus wine was mainly related to the content of ester, terpenes, aldehydes and ketones. Low content of terpenes and aldehydes and ketones also contributed positively to the sensory quality of citrus wine. Generally, many aroma compounds in wine will have a synergistic effect on its flavor, an unbalanced aroma compounds level would have adverse effects on the sensory quality of wine.

In addition, biocontrol strategies for the limitation of undesired microbial growth in foods and beverages by the application of non-*Saccharomyces* yeasts have been highlighted in recent years [66,67]. Berbegal et al. [68] reported that the use of a mixed starter of different strains of *S. cerevisiae* or a mix of specific strains of non-*Saccharomyces* yeast with *S. cerevisiae* are efficient strategies in the control of the spoilage yeast like *Brettanomyces bruxellensis* and the volatile phenols’ production in wine. Interestingly, the inoculation of different non-*Saccharomyces* and the inoculation time of the non-*Saccharomyces* yeast with respect to *S. cerevisiae* resources (co-inoculated and sequentially inoculated) influence the composition of the connected malolactic fermentation consortia, modulating malolactic fermentation performance [69].

## 4. Conclusions

The selected non-*Saccharomyces* yeast strains and *S. cerevisiae* could grow normally during their co-fermentations with the exception of *H. uvarum,* which was inhibited by *S. cerevisiae*, and produced citrus wines with low amount of ethanol and total acidity, and high residual sugar content and pH value, and with no significant difference in volatile acid concentration compared with those of *S. cerevisiae* fermentation. The co-fermentations produced citrus wine with higher content of volatile aroma compounds, especially higher alcohols and esters, than pure *S. cerevisiae* fermentation did. However, they had no regular effects on the content of fatty acids, terpenes, aldehydes, ketones, and phenols. At the same time, different fermentations including co-fermentations and pure fermentation, produced different aroma compound profiles, and odor active compound profiles, and would present on different flavor complexities and characteristics in citrus wines. More importantly, all the co-fermentations could improve the sensory quality of citrus wine, Sc-Hop co-fermentation produced the best sensory quality of citrus wine, followed by Sc-Hu and Sc-Td co-fermentations. All the results indicated these co-fermentations, producing high content of volatile aroma compounds, especially higher alcohols and esters, benefitted the sensory quality improvement of citrus wine. However, the mechanism that co-fermentation can increase the content of higher alcohols in citrus wine is not known.

## Figures and Tables

**Figure 1 microorganisms-08-00323-f001:**
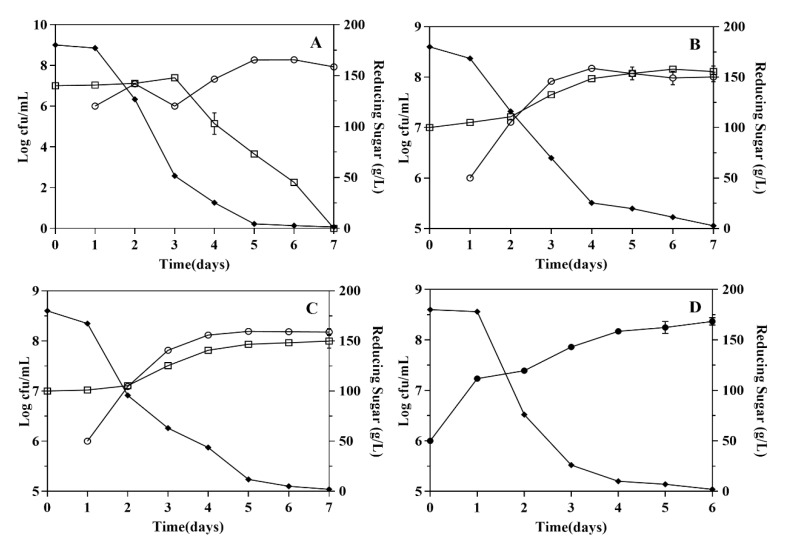
Growth kinetics and sugar consumption kinetics of yeast strains during fermentations. (**A**, **B** and **C** are co-fermentation with *S. cerevisiae*, and **D** is pure culture). **A**. *H. uvarum*; **B**. *T. delbrueckii*; **C**. *H. opuntiae*; **D**. *S. cerevisiae*. -□- Growth kinetics of non-*Saccharomyces* in co-fermentations -♦- Sugar consumption kinetics. -ο- Growth kinetics of *S. cerevisiae* in co-fermentations -●- Growth kinetics of *S. cerevisiae* in pure fermentation

**Figure 2 microorganisms-08-00323-f002:**
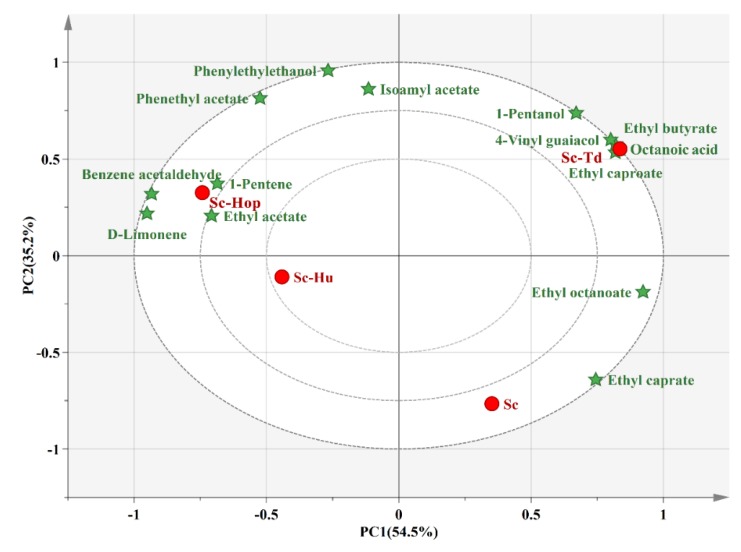
Principal component analysis of volatile aroma compounds in citrus wines. Sc-Hu, *S. cerevisiae*/*H. uvarum* co-fermentation; Sc-Td, *S. cerevisiae*/*T. delbrueckii* co-fermentation; Sc-Hop, *S. cerevisiae*/*H. opuntiae* co-fermentation; Sc, *S. cerevisiae* fermentation.

**Figure 3 microorganisms-08-00323-f003:**
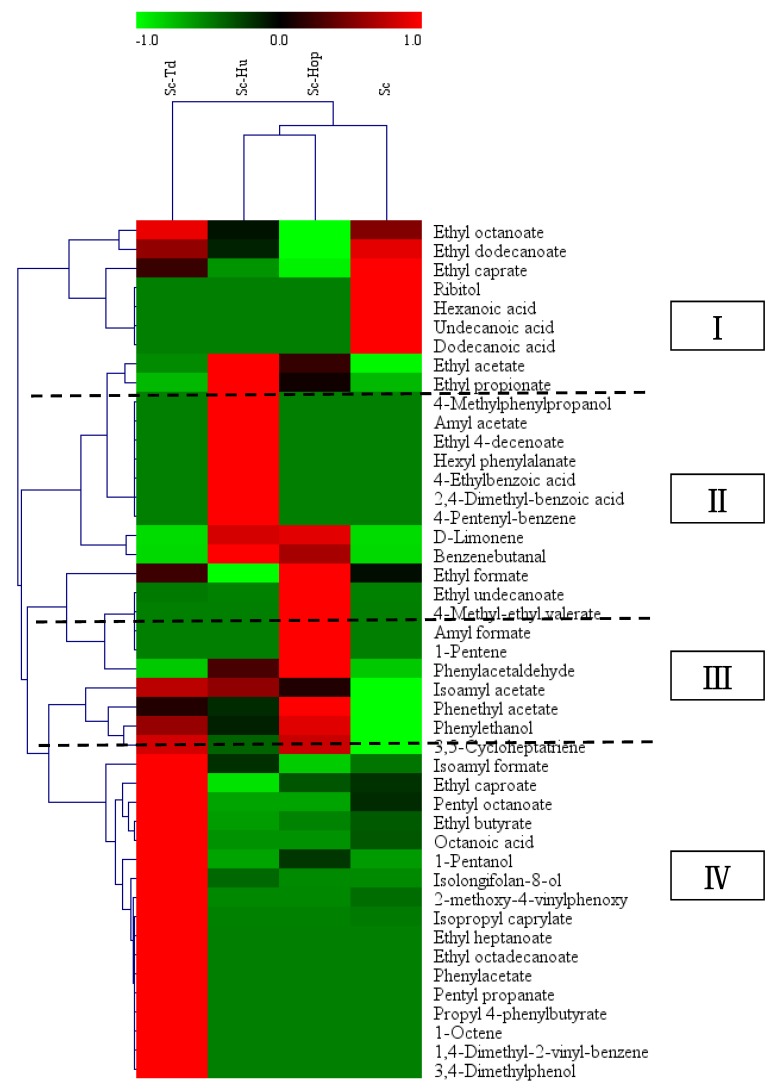
Hierarchical cluster analysis of volatile compounds in citrus wines. Normalization of all values for better visualization; Sc-Hu, *S. cerevisiae*/*H. uvarum* co-fermentation; Sc-Td, *S. cerevisiae*/*T. delbrueckii* co-fermentation; Sc-Hop, *S. cerevisiae*/*H. opuntiae* co-fermentation; Sc, *S. cerevisiae* fermentation.

**Table 1 microorganisms-08-00323-t001:** Physicochemical parameters of citrus wines (Means ± SD).

Fermentations	Residual Sugar	Ethanol	Total Acidity ^a^	Volatile Acid ^b^	pH
	(g/L)	(%, v/v)	(g/L)	(g/L)	
**Sc-Hu**	1.38 ± 0.08 **	10.24 ± 0.08 *	7.68 ± 0.01 **	0.12 ± 0.00	3.45 ± 0.00 **
**Sc-Td**	2.81 ± 0.13 **	9.74 ± 0.12 **	8.63 ± 0.06 **	0.15 ± 0.01 **	3.47 ± 0.01 **
**Sc-Hop**	2.03 ± 0.08	9.83 ± 0.49 *	8.52 ± 0.02 **	0.11 ± 0.02	3.37 ± 0.06
**Sc**	2.03 ± 0.13	11.29 ± 0.26	9.44 ± 0.07	0.11 ± 0.01	3.35 ± 0.00

Data show the mean value of triplicate, and the significant differences in each row are respect to the control Sc. ^a^ Expressed as g/L of malic acid; ^b^ Expressed as g/L of acetic acid ** *p* < 0.01; * *p* < 0.05. Sc-Hu, *S. cerevisiae/H. uvarum* co-fermentation; Sc-Td, *S. cerevisiae/T. delbrueckii* co-fermentation; Sc-Hop, *S. cerevisiae/H. opuntiae* co-fermentation; Sc, *S. cerevisiae* fermentation.

**Table 2 microorganisms-08-00323-t002:** Volatile aroma compounds in citrus wines (Means ± SD, mg/L).

Number	Compounds	Sc-Hu	Sc-Td	Sc-Hop	Sc
	**Higher alcohols**				
**1**	1-Pentanol	49.61 ± 0.39 ^a^	309.54 ± 1.47 ^b^	103.00 ± 1.00 ^c^	53.95 ± 0.29 ^d^
**2**	Isohexanol	0.84 ± 0.04	-	-	-
**3**	Ribitol	-	-	-	0.81 ± 0.02
**4**	3-Methyl-1-hexanol	0.76 ± 0.04	-	-	-
**5**	4-Methylphenylpropanol	0.06 ± 0.01 ^a^	-	0.21 ± 0.02 ^b^	-
**6**	Neopentyl glycol	1.28 ± 0.04 ^a^	-	0.99 ± 0.03 ^b^	-
**7**	Phenylethanol	183.00 ± 3.00 ^a^	270.99 ± 3.01 ^b^	306.33 ± 4.51 ^c^	33.21 ± 0.04 ^d^
**8**	Isolongifolan-8-ol	0.02 ± 0.01^a^	0.31 ± 0.03 ^b^	-	-
	***subtotal***	**235.52 ± 3.48 ^a^**	**580.72 ± 4.34 ^b^**	**410.48 ± 5.39 ^c^**	**87.97 ± 0.24 ^d^**
	**Fatty acids**				
**1**	Dodecanoic acid	-	-	-	0.04 ± 0.01
**2**	Hexanoic acid	-	-	-	0.13 ± 0.03
**3**	Octanoic acid	-	20.04 ± 0.84 ^a^	-	2.23 ± 0.05 ^b^
**4**	Undecanoic acid	-	-	-	0.15 ± 0.02
**5**	4-Ethylbenzoic acid	0.10 ± 0.02	-	-	-
**6**	2,4-Dimethyl-benzoic acid	0.06 ± 0.01	-	-	-
	***subtotal***	**0.16 ± 0.03 ^a^**	**20.04 ± 0.84 ^b^**	**0.00 ± 0.00**	**2.54 ± 0.12 ^c^**
	**Acetates**				
**1**	Ethyl acetate	153.50 ± 3.50 ^a^	32.04 ± 0.74 ^b^	81.64 ± 0.66 ^c^	4.36 ± 0.10 ^d^
**2**	Isoamyl acetate	45.41 ± 0.81 ^a^	47.60 ± 0.40 ^b^	39.53 ± 0.53 ^c^	18.51 ± 0.49 ^d^
**3**	Amyl acetate	0.19 ± 0.01	-	-	-
**4**	Phenethyl acetate	37.46 ± 0.76 ^a^	46.59 ± 0.51 ^b^	78.55 ± 0.45 ^c^	6.94 ± 0.06 ^d^
	***subtotal***	**236.82 ± 2.18 ^a^**	**126.36 ± 0.44 ^b^**	**199.59 ± 0.53 ^c^**	**29.81 ± 0.52 ^d^**
	**Ethyl esters**				
**1**	Ethyl butyrate	0.34 ± 0.04 ^a^	4.93 ± 0.02 ^b^	0.57 ± 0.02 ^c^	0.91 ± 0.02 ^d^
**2**	Ethyl caprate	14.39 ± 0.39 ^a^	19.12 ± 0.48 ^b^	12.25 ± 0.05 ^c^	25.38 ± 0.62 ^d^
**3**	Ethyl caproate	6.45 ± 0.35 ^a^	12.40 ± 0.30 ^b^	7.87 ± 0.06 ^c^	8.23 ± 0.05 ^d^
**4**	Ethyl dodecanoate	0.38 ± 0.02 ^a^	0.60 ± 0.03 ^b^	-	0.70 ± 0.02 ^c^
**5**	Ethyl formate	-	0.19 ± 0.03 ^a^	0.29 ± 0.01 ^b^	0.15 ± 0.01 ^c^
**6**	Ethyl heptanoate	-	0.81 ± 0.02	-	-
**7**	Ethyl octanoate	21.73 ± 0.42 ^a^	37.81 ± 0.70 ^b^	1.01 ± 0.01 ^c^	31.26 ± 0.74 ^d^
**8**	Ethyl octadecanoate	-	0.10 ± 0.01	-	-
**9**	Ethyl propionate	0.51 ± 0.02 ^a^	-	0.19 ± 0.01 ^b^	-
**10**	Ethyl undecanoate	-	0.86 ± 0.05 ^a^	51.04 ± 0.16 ^b^	-
**11**	Ethyl 4-methyl-valerate	-	-	0.09 ± 0.01	-
**12**	Ethyl cis 4-decenoate	0.22 ± 0.03	-	-	-
	***Subtotal***	**43.99 ± 1.28 ^a^**	**77.17 ± 1.41 ^b^**	**73.31 ± 0.29 ^c^**	**66.62 ± 1.43 ^d^**
	**Other esters**				
**1**	Amyl formate	-	-	0.77 ± 0.01	-
**2**	Hexyl phenylalanate	0.25 ± 0.04	-	-	-
**3**	Isoamyl formate	0.55 ± 0.35 ^a,b^	2.01 ± 0.10 ^c^	-	0.31 ± 0.02 ^b^
**4**	Isopropyl caprylate	-	1.94 ± 0.05 ^a^	-	0.03 ± 0.01 ^b^
**5**	Pentyl propanate	-	2.48 ± 0.12	-	-
**6**	Pentyl octanoate	-	0.31 ± 0.02 ^a^	-	0.07 ± 0.01 ^b^
**7**	Propyl 4-phenylbutyrate	-	0.26 ± 0.03	-	-
**8**	Phenylethyl phenylacetate	-	0.35 ± 0.04	-	-
	***subtotal***	**0.80 ± 0.32 ^a^**	**6.99 ± 0.27 ^b^**	**0.77 ± 0.01 ^a^**	**0.41 ± 0.04 ^c^**
	***total esters***	**281.61 ± 3.78 ^a^**	**210.52 ± 2.12 ^b^**	**273.67 ± 0.83 ^c^**	**96.84 ± 1.99 ^d^**
	**Terpenes**				
**1**	1-Octene	-	12.12 ± 9.98	-	-
**2**	1-Pentene	-	-	2.77 ± 0.03	-
**3**	1,4-Dimethyl-2-vinyl-benzene	-	0.26 ± 0.01b	-	-
**4**	3,5-Cycloheptatriene	0.25 ± 0.04 ^a^	0.60 ± 0.06 ^b^	0.59 ± 0.01 ^b^	-
**5**	4-Pentenyl-benzene	0.65 ± 0.01	-	-	-
**6**	D-Limonene	1.22 ± 0.22 ^a^	-	1.26 ± 0.04 ^a^	-
	***subtotal***	**2.12 ± 0.25 ^a^**	**12.98 ± 0.04 ^b^**	**4.62 ± 0.02 ^c^**	**0.00 ± 0.00**
	**Aldehydes and ketones**				
**1**	Phenylacetaldehyde	0.10 ± 0.08 ^a^	-	0.19 ± 0.02 ^b^	-
**2**	Benzenebutanal	0.46 ± 0.03 ^a^	-	0.36 ± 0.01 ^b^	-
	***subtotal***	**0.56 ± 0.11 ^a^**	**0.00 ± 0.00**	**0.55 ± 0.03 ^a^**	**0.00 ± 0.00**
	**Phenols**				
**1**	4-Vinyl guaiacol	-	4.42 ± 0.09 ^a^	-	0.23 ± 0.03 ^b^
**2**	3,4-Dimethylphenol	-	0.79 ± 0.03	-	-
	***subtotal***	**0.00 ± 0.00**	**5.20 ± 0.12 ^a^**	**0.00 ± 0.00**	**0.23 ± 0.03 ^b^**
	***Total***	**519.97 ± 7.65 ^a^**	**819.46 ± 7.46 ^b^**	**689.32 ± 6.27 ^c^**	**187.58 ± 2.38 ^d^**

Data show the mean value of triplicate, and “-” presented no detection; Different letters in each row presented statistically significant differences among citrus wines at *p* < 0.05. Sc-Hu, *S. cerevisiae/H. uvarum* co-fermentation; Sc-Td, *S. cerevisiae/T. delbrueckii* co-fermentation; Sc-Hop, *S. cerevisiae/H. opuntiae* co-fermentation; Sc, *S. cerevisiae* fermentation. Bold value indicated the subtotal value of the type of aroma compounds

**Table 3 microorganisms-08-00323-t003:** Volatile odor-active compounds (OAV ≥ 1) in citrus wines.

Compounds	Description	Threshold ^a^	Sc-Hu	Sc-Td	Sc-Hop	Sc
1-Pentanol	Fruity, balsamic	8 [41]	6.20	38.70	12.88	6.74
Phenylethanol	Honey, rose, spicy	14 [42]	13.07	19.36	21.88	2.37
Ethyl acetate	Pineapple, fruity, solvent	7.5 [42]	20.47	4.27	10.89	0.58
Isoamyl acetate	Banana, pear	0.03 [42]	1513.67	1586.67	1317.67	617.00
Phenethyl acetate	Rose, honey, tobacco	0.25 [42]	149.84	186.36	314.20	27.76
Ethyl butyrate	Strawberry	0.02 [43]	17.00	246.25	28.25	45.25
Ethyl caproate	Pineapple, fruity, flowery	0.014 [44]	460.71	885.71	561.79	587.86
Ethyl octanoate	Pineapple, pear, soapy	0.002 [42]	10,862.50	18,902.50	505.00	15,627.50
Ethyl caprate	Fruity, fresh	0.28	51.39	68.29	43.75	90.64
Octanoic acid	Sweat, cheesy	0.5 [43]	0.00	40.08	0.00	4.45
D-Limonene	Floral, green, citrus	0.2 [41]	6.10	0.00	6.30	0.00
1-Pentene	Honey, peach, sweat	0.02678 [45]	0.00	0.00	103.44	0.00
Phenylacetaldehyde	Rose	0.001 [46]	100.00	0.00	193.33	0.00
4-Vinyl guaiacol	Coffee, beer, apple	1.1 [42]	0.00	4.01	0.00	0.21

^a^ Threshold values were expressed as mg/L. OVA was the ratio between concentration of the volatile compound to its odor threshold. Sc-Hu, *S. cerevisiae/H. uvarum* co-fermentation; Sc-Td, *S. cerevisiae/T. delbrueckii* co-fermentation; Sc-Hop, *S. cerevisiae/H. opuntiae* co-fermentation; Sc, *S. cerevisiae* fermentation.

**Table 4 microorganisms-08-00323-t004:** Sensory evaluation scores of citrus wines.

Fermentations	Clarity	Aroma (Fruity, Floral)	Taste	Taste Lasting	Overall Acceptability	Total
	/3	/6	/6	/3	/2	/20
**Sc-Hu**	2.83 ± 0.41 **	4.16 ± 0.41 **	3.83 ± 0.41 **	1.83 ± 0.41 **	1.33 ± 0.52 *	14.00 ± 0.63 **
**Sc-Td**	2.16 ± 0.41 **	3.67 ± 0.52 **	3.83 ± 0.41 **	2.00 ± 0.00 **	1.33 ± 0.52 *	13.00 ± 0.63 **
**Sc-Hop**	2.5 ± 0.55 **	4.5 ± 0.55 **	4.33 ± 0.52 **	2.00 ± 0.00 **	1.67 ± 0.52 **	15.00 ± 0.67 **
**Sc**	1.50 ± 0.55	2.83 ± 0.41	2.67 ± 0.52	1.00 ± 0.00	1.00 ± 0.00	9.00 ± 0.89

Data show the mean value of the scores from 9 panelists, and the significant differences were with respect to the control Sc. ** *p* < 0.01; * *p* < 0.05. Sc-Hu, *S. cerevisiae/H. uvarum* co-fermentation; Sc-Td, *S. cerevisiae/T. delbrueckii* co-fermentation; Sc-Hop, *S. cerevisiae/H. opuntiae* co-fermentation; Sc, *S. cerevisiae* fermentation.

**Table 5 microorganisms-08-00323-t005:** Correlation analysis between sensory evaluation scores and aroma compounds.

Compounds Categories	Clarity	Aroma	Taste	Taste Lasting	Acceptability	Total
**Higher alcohols**	0.307	0.333	0.518	**0.751 ****	0.305	0.469
**Total esters**	**0.861 ****	**0.863 ****	**0.94 ****	**0.928 ****	**0.795 ****	**0.952 ****
**Fatty acids**	−0.196	−0.265	−0.087	−0.217	−0.246	−0.141
**Terpenes**	0.561	**0.66 ***	**0.792 ****	**0.84 ****	**0.642 ***	**0.754 ****
**Aldehydes and ketones**	**0.709****	**0.748 ****	**0.769 ****	0.57	**0.775 ****	**0.778 ****
**Phenols**	−0.128	−0.194	−0.01	0.293	−0.18	−0.064

Bold value indicated significant different, and *, ** indicated significant difference at 5% and 1%, respectively.

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
