# Peer review of "The Sensory Quality Improvement of Citrus Wine through Co-Fermentations with Selected Non-Saccharomyces Yeast Strains and Saccharomyces cerevisiae"

_microorganisms, 2020, doi:10.3390/microorganisms8030323_

Round 1
Reviewer 1 Report
In my opinion the manuscript is well structured. I propose some indications:
Line 118 Results and Discussion
The discussion is poor.I suggest to better discuss the data obtained, providing explanations on the formation of the volatile molecules identified on the basis of the literature; The authors studied the volatile component through 3 approaches: qualitative-quantitative determination of VOCs, evaluation of OAV and sensory approach. It would be interesting to complete with the correlation between analytical and sensory data using multivariate techniques.
Reviewer 2 Report
In this manuscript authors analyze the effect of mixed fermentations with selected non-Saccharomyces yeast and Saccharomyces cerevisiae on the fermentation performance and flavor of citrus wine. They compare the data obtained by a control strain of S.cerevisiae, with its use on sequential fermentations with H.uvarum, T. delbrueckii and H. opuntiae. Authors compare sugar consumption, yeast growth, and physicochemical analysis of citrus wines, including the different volatile aroma compounds. Authors also include a sensory evaluation of the obtained citrus wines.
Even though the aim of the manuscript is well addressed, as similar studies had been done in the production of other fermented beverages, such as wine, the discussion of the results can be improved, emphasizing the impact of the non-Saccharomyces species used on the citrus wine, and which is the expected contribution to practical application. The introduction could also be extended, with more references to previous studies performed with mixed inoculum, and the effect observed with the different species used in this study. Overall, the study presents some interesting results, and generally well written, however, some issues need to be addressed.
Title and abstract: Authors refer in the title, abstract and conclusions of the manuscript to the quality of the citrus wine. How do authors define or measure the quality of a citrus wine? Which compounds or aromas should be increased or decreased? The word quality sounds ambiguous and subjective, as it cannot be measured or compared. Define how the quality is measured (parameters directly related to it), or rephrase.
Section 2.1. Which kind of citric fruit are authors using? Please indicate. Which are the characteristics of the citrus juice? (sugar content, pH, etc)
Section 2.3.
Line 80. Rewrite as follows “After adjusting the final concentration of sugar and SO2 of citrus juice to…” and indicate how those parameters were adjusted, in which form and amount the sugar and SO2 were added.
Lines 84-86: Rewrite: “The co-fermentations were performed by inoculating 107 cell/ml of the corresponding non-Saccharomyces yeast strain, and 106 cell/ml of S.cerevisiae after 24 h of H.opuntiae and T.delbrueckii inoculation, or after 72 h of H.uvarum inoculation.
Line 89-90. Name of the method used for calculating residual sugars concentration.
Section 2.4: Name the methods used for the physicochemicalanalysis of wines, not only referring to OIV.
Section 2.5, lines 104-105: include the reference/s for the odor threshold of the compounds. Line 139: refer to total acidity, not total acid.
Section 3.1. Figure 1: indicate in the figure caption or legend that A, B and C are co-fermentation with S. cerevisiae, and D is pure culture. For Sugar consumption, same legend/symbol can be used for all the graphs (no need to differentiate co-fermentation from pure)
Section 3.2. Line 151: higher residual sugar content was obtained only with Td, not all the co-fermentation. Rephrase.
Table 1. Indicate that the significant differences are respect to the control Sc
Section 3.3. More references and discussion would be needed, mainly in sections 3.3.1, 3.3.3, 3.3.4. Authors would need to include some more discussion of the results, comparing with previous studies or known facts. In some parts they mainly describe the results obtained, without extracting the relevance of them and the outcome for the final product. For example, when describing the different volatile compounds produced, they indicate that excessive concentration of higher alcohols might result in undesirable flavor (line 181). Which would be excessive? Reference? However, they see an increase on phenyl ethanol, which aroma is not considered unpleasant, thus, in that case higher amounts would be positive. On the other hand, when describing the ester compounds, authors mainly observe an increase on the ethyl acetate, which odor can be negative in wines (nail polish, solvent,etc). What about the levels found in citrus wines? Could negatively affect the aroma? Comment and discuss about it. Similar studies had been done in wine production (from grape must), using the same species in co-inoculation. Authors would need to compare their results with those, to see if the contribution or modification on the aroma content of the final product is species specific, or how much the media/must used affects to the aroma modulation.
Line 162: replace “high content” by “higher content”.
Line 166: Sc-Td condition produced the highest amount of volatile fatty acids (around 20mg/L), but authors indicate that co-fermentations produced lower content. Rephrase.
Line 178: replace “higher alcohol” by “higher alcohols”
Table 3: Clarify the table legend, explaining the OAV values. The references included as footnotes of Table 3 are not included in the reference section.
Section 3.4. – Figure 2. More detail should be given about the PCA analysis. What variables are included in each component? In figure 2 some names are overlapping. Correct.
Section 3.5. – Figure 3. Values were normalized to what? To the highest value? Explain and clarify. The non-detected compounds should be shown in another color, as green is too similar to the heat map and creates confusion. Even if the heat map and clustering is a visual way of showing the main differences among volatile compounds, as a lot of the compounds were only detected in one of the conditions, it makes the analysis uncertain.
Section 3.6. – The items analyzed in the sensory analysis are too broad and subjective: what appearance/aroma/taste means? How to measure good or bad? For a better objectivity of the analysis, more detailed concepts should be used: e.g fruity aroma, acidity, turbidity, etc. The best test to measure objective statistical differences on sensory analysis is the triangular test, which indicates if there are differences or not with the control. How those results agree or disagree with the aroma production?
Table 4 indicates that data show the mean value of triplicates, however, according section 2.6, 9 panelists were doing the sensory evaluation. Are the values of Table 4 the mean of the 9 panelists? Indicate that the significant differences were with respect to the control Sc.
Conclusions - Lines 315-318: what indicates the best quality of a citrus wine? The preference among the sensory analysis? Could that relate to the results obtained on the volatile content analysis?
References: write the name of species in italics.
